# Sarcopenic Obesity Is a Risk Factor for Worse Oncological Long-Term Outcome in Locally Advanced Rectal Cancer Patients: A Retrospective Single-Center Cohort Study

**DOI:** 10.3390/nu15112632

**Published:** 2023-06-05

**Authors:** Peter Tschann, Markus P. Weigl, Patrick Clemens, Philipp Szeverinski, Christian Attenberger, Matthias Kowatsch, Tarkan Jäger, Klaus Emmanuel, Thomas Brock, Ingmar Königsrainer

**Affiliations:** 1Department of General- and Thoracic Surgery, Academic Teaching Hospital, 6800 Feldkirch, Austria; markus.weigl@lkhf.at (M.P.W.); thomas.brock@lkhf.at (T.B.); ingmar.koenigsrainer@lkhf.at (I.K.); 2Department of Radio-Oncology, Academic Teaching Hospital, 6800 Feldkirch, Austria; patrick.clemens@lkhf.at; 3Institute of Medical Physics, Academic Teaching Hospital, 6800 Feldkirch, Austria; philipp.szeverinski@lkhf.at (P.S.); christian.attenberger@lkhf.at (C.A.); matthias.kowatsch@lkhf.at (M.K.); 4Department of Surgery, Paracelsus Medical University Salzburg, 5020 Salzburg, Austria; ta.jaeger@salk.at (T.J.); k.emmanuel@salk.at (K.E.)

**Keywords:** body composition, sarcopenia, sarcopenic obesity, locally advanced rectal cancer, morbidity, oncological long-term outcome

## Abstract

Background: Malnutrition and skeletal muscle waste (sarcopenia) are known as predictive factors for a poor postoperative outcome. Paradoxically, obesity seems to be associated with a survival advantage in wasting diseases such as cancer. Thus, the interpretation of body composition indices and their impact on rectal cancer therapy has become more and more complex. The aim of this study was to evaluate body composition indices in locally advanced rectal cancer patients prior to therapy and their impact on short- and long-term outcomes. Methods: Between 2008 and 2018, 96 patients were included in this study. Pre-therapeutic CT scans were used to evaluate visceral and subcutaneous fat mass, as well as muscle mass. Body composition indices were compared to body mass index, morbidity, anastomotic leakage rate, local recurrency rate, and oncological long-term outcomes. Results: Increased visceral fat (*p* < 0.01), subcutaneous fat (*p* < 0.01), and total fat mass (*p* = 0.001) were associated with overweight. Skeletal muscle waste (sarcopenia) (*p* = 0.045), age (*p* = 0.004), comorbidities (*p* < 0.01), and sarcopenic obesity (*p* = 0.02) were significantly associated with increased overall morbidity. The anastomotic leakage rate was significantly influenced when comorbidities were present (*p* = 0.006). Patients with sarcopenic obesity showed significantly worse disease-free (*p* = 0.04) and overall survival (*p* = 0.0019). The local recurrency rate was not influenced by body composition indices. Conclusion: Muscle waste, older age, and comorbidities were demonstrated as strong risk factors for increased overall morbidity. Sarcopenic obesity was associated with worse DFS and OS. This study underlines the role of nutrition and appropriate physical activity prior to therapy.

## 1. Introduction

Colorectal cancer (CRC) is the third leading cause of cancer death worldwide, and represents 10.2% of all new cancer diagnoses [1,2]. Rectal cancer diagnoses account for 730 thousand of 1.9 million new colorectal cancer cases [3]. Multidisciplinary therapy, including preoperative radiation or combined chemo-radiation therapy followed by surgery, remains the cornerstone in non-metastatic locally advanced rectal cancer patients [4]. Depending on tumor size, tumor location, and stage of disease, a sphincter-sparing procedure with primary anastomosis is the most often performed. As an alternative to a low anterior resection (LAR) with primary anastomosis, an end-colostomy can be performed in case of the patient’s wish or in patients with a higher risk of anastomotic leakage (AL). Each option for rectal cancer surgery bears a risk of complications. However, AL is the most relevant complication after LAR with primary anastomosis. The rate of AL has been shown to differ between 1 and 30% in the previously published literature [4,5,6,7,8,9]. Severe complications, especially AL, are associated with increased morbidity, longer hospitalization, higher costs, and possibly a worse oncological outcome [10,11].

Malnutrition and skeletal muscle depletion (sarcopenia) are known as predictive factors for worse postoperative outcomes and poor long-term survival in patients diagnosed with colorectal cancer [12,13,14]. However, our understanding of how to interpret human body weight and its impact on cancer therapy has become more complex recently. Obesity is widely known as a risk factor for cardiovascular diseases and is, therefore, associated with a decreased life expectancy [15]. Paradoxically, obesity diagnosed in terms of body mass index (BMI) seems to be associated with a survival advantage in wasting diseases, including cancer, rather than a disadvantage [14]. However, aside from technical difficulties in the performance of open and laparoscopic colorectal resections in obese patients, visceral fat is associated with a higher rate of intra- and postoperative morbidity [4,16,17]. Moreover, the risk of inadequate mesorectal excision is reported to be higher in obese patients, with a possible impaired oncological outcome [16,18]. Nevertheless, recently published literature shows no evidence of a worse oncological outcome in obese rectal cancer patients [16,19]. 

Most of the previous published literature used BMI as a descriptive value for the definition of obesity. Literature regarding more specific parameters for the evaluation of obesity or malnutrition and its impact on short- and long-term outcome after therapy for locally advanced rectal cancer is rare, but essential, to differentiate between obese patients with and without sarcopenia. The aim of this study was to evaluate the impact of visceral fat and body composition in locally advanced rectal cancer patients undergoing neoadjuvant therapy followed by surgery, primarily on postoperative morbidity and mortality, and secondarily on the long-term oncological outcome. 

## 2. Methods

### 2.1. Patients and Eligibility

After institutional review board approval and in accordance with the Ethics Committee of the Province of Vorarlberg (EK-0.04-440), data were retrieved from a prospective maintained database of the Academic Teaching Hospital in Feldkirch. From January 2008 to December 2018, all locally advanced rectal cancer patients (cT3, cT4, N +) who received neoadjuvant therapy followed by surgery were included in this study. Exclusion criteria were defined as follows: loss of follow-up, metastatic disease, abdominoperineal resections, and computer tomography (CT) scan pictures being unavailable for evaluation. 

### 2.2. Tumor Assessment

All patients received a colonoscopy with tissue biopsy. A CT scan of the trunk was assessed to rule out metastatic disease. A pelvic magnet resonance imaging (MRI) scan and an endorectal ultrasound were performed to assess local tumor staging. Tumor height from the anal verge was analyzed by proctoscopy. 

### 2.3. Baseline Variables

The baseline patients’ variables included sex, age, BMI, ASA classification [20], preoperative tumor staging, type of preoperative therapy, type of anastomosis, preoperative carcinoembryonic antigen level, postoperative pathological stage, tumor regression, duration of the hospital stay, postoperative complications (according to the Dindo–Clavien Classification [21]), local recurrency, and distant metastasis. Clinical and pathological staging were based on the 8th edition of the Union for International Cancer Control (UICC) Classification of malignant tumors [22]. Local recurrency was defined as any tumor recurrency in the pelvic cavity that was confirmed by radiological or histological examination. Disease-free survival (DFS) was defined as months from the date of surgery to the date of the detection of either local recurrency, distant metastasis, last follow-up, or death. Overall survival (OS) was defined from the date of surgery to the date of death. 

### 2.4. Treatment Strategy

All locally advanced rectal cancer patients were individually discussed in a multidisciplinary team discussion, and neoadjuvant therapy was indicated in accordance with international guidelines. Neoadjuvant therapy was either performed as short-term radiation therapy (5 Gy per day for 5 days = 5 × 5 Gy) or as combined chemo-radiation therapy (5-Fluouracil or Capecitabine plus 50.4 Gy). In the case of short-term radiation therapy, surgery was performed within 7 days, while in the case of combined chemo-radiation therapy, the resection was usually performed 8 weeks after the end of preoperative therapy. 

### 2.5. Histopathological Examination

The removed tissue was immediately fixed with formalin. The pathological examination included a macroscopic description of the removed tissue, specification of the circumferential resection margin and distal resection margin, and complete histopathological staging. The assessment of the total mesorectal excision (TME) quality was modified from the score established by Phil Quirke, (good = 1, moderate = 2, poor = 3) [23,24]. 

### 2.6. CT Evaluation

Staging CT scans taken fewer than 30 days prior to elective surgery were utilized to determine the body composition parameters. The CT scans were retrieved from the imaging software Deep Unity Diagnost (DH Healthcare GmbH, Version 1.2.0.1). The muscle and fat areas were evaluated at the level of the umbilicus and exported to a 3D visualization program (Horos^TM^, v3.3.6) for further analysis (Figure 1). For skeletal muscle tissue, Hounsfield unit (HU) thresholds of −30 to +110 were used, while the thresholds for visceral and subcutaneous fat were −190 HU to −30 HU. To determine the impact of different fat distributions, the following parameters were examined: total fat area (TFA, cm^2^), visceral fat area (VFA, cm^2^), subcutaneous fat area (SFA, cm^2^), the ratio of visceral to total fat area (VFA/TFA), and the ratio of subcutaneous to total fat area (SFA/TFA). Gender-based cut offs were used to assess increased values of the mentioned parameters [25]. To further evaluate the muscle mass of the included patients, the overall skeletal area (SMA, cm^2^) and the height-adjusted skeletal muscle index (SMI, cm^2^/m^2^) were applied. Two different definitions were used to identify sarcopenia. On the one hand, an SMA or SMI less than two standard deviations below the mean was set as the cut-off for sarcopenic patients. On the other hand, a gender-specific standardized value (SMA/SMI ≤ 5th percentile) was applied [26]. Sarcopenic obesity was defined as a low SMA and a high visceral fat level. Patients with a BMI >25 kg/m^2^ were defined as overweight, and those with a BMI <25 kg/m^2^ as underweight. Cut-off values are shown in Appendix A.

### 2.7. Statistical Analysis

Statistical analysis was carried out using the programming language R (Version 4.2.2). Continuous data on the patients’ characteristics were tested for normal distribution using the Shapiro–Wilk test [27]. Normally distributed data are presented as mean ± standard deviation. Continuous data were assessed by either the *t*-test, the Mann–Whitney U-test, or the Kruskal–Wallis Test. Categorical data are presented in absolute numbers (percent), and were assessed using the Chi-square test. The survival analysis was conducted using Kaplan–Meier curves to graphically show the OS and DFS grouped for sarcopenic obesity and non-sarcopenic obesity. To compare these groups, a log rank test was utilized. The median follow-up time was calculated using the inverse Kaplan–Meier method. Statistical power was calculated with significance set at a *p*-value of <0.05.

## 3. Results

Patients’ characteristics, perioperative therapy courses, and postoperative outcomes are shown in Table 1. The study group consisted of 64 male (66.7%) and 32 female (33.3%) patients, with an average age of 64 years (±11.0) and a mean BMI of 26.7 kg/m^2^ (±4.2). In total, 14 (14.6%) patients were ASA I, 49 (51.0%) patients were ASA II, 30 patients (31.3%) were ASA III, and 1 patient had an ASA score of IV (1.0%). One or more comorbidities were observed in 49 cases (51.0%) prior to surgery, and the average Charlson Comorbidity Index (CCI) was 5.4 (±2.5). Overall morbidity was 37.5%. An anastomotic leakage occurred in 20 patients (20.8%). The mean duration of the hospital stay was 20.0 days (±14.2). In the majority of evaluated patients, the TME quality was good (69.8%). The mean follow-up time was 69.0 months (±46.5).

The association between BMI and body composition indices is shown in Table 2. Patients with increased subcutaneous adiposity (*p* < 0.01), visceral adiposity (*p* < 0.01), and high total fat (*p* = 0.001) were significantly associated with overweight. The visceral-to-total fat ratio (VF/TF ratio) and subcutaneous-to-total fat ratio (SF/TF ratio) moderately correlated with overweight (VF/TF ratio: *p* = 0.058; SF/TF ratio: *p* = 0.06).

The association between the baseline and therapeutical characteristics, as well as the body composition indices, is shown in Table 3. Overall morbidity was significantly affected by older age (*p* = 0.004), increased CCI (*p* < 0.01), lower SMA (*p* = 0.045), and sarcopenic obesity (*p* = 0.02). A low skeletal muscle index (*p* = 0.053) and a higher SF/TF ratio (*p* = 0.096) moderately correlated with increased overall morbidity. The anastomotic leakage rate was significantly affected by increased CCI (*p* = 0.006). The local recurrency rate was not affected by the analyzed parameters nor was DFS when categorical numbers were compared. However, DFS and OS were significantly influenced by sarcopenic obesity in the Kaplan–Meier survival curves over time (*p* = 0.04/*p* = 0.0019). Disease-free and overall survival curves are shown in Figure 2. Other body composition indexes had no influence on DFS or OS. DFS and overall survival curves of BMI, SF, VF and TF are shown in Appendix A.

In our study assessing the differences in sarcopenic obesity between individuals with and without overall morbidity, we performed a power analysis. We found that with our sample sizes of 35 in the case group and 61 in the control group, and an observed effect size based on sarcopenic obesity rates of approximately 22.9% and 4.9% in the case and control groups, respectively, our test had a power of approximately 73.6%.

## 4. Discussion

Malnutrition and sarcopenia are suggested to enhance morbidity and mortality rates in patients with locally advanced rectal cancer undergoing neoadjuvant therapy followed by surgery. In the present study, we evaluated body composition indices in association with morbidity, anastomotic leakage rate, and oncological outcome after multimodal therapy. The data evaluation suggests that sarcopenia is associated with increased morbidity after TME. Moreover, sarcopenic obesity is significantly associated with impaired disease-free and overall survival. 

Malnutrition is well-known as a risk factor for postoperative complications and worse oncological outcomes [28]. The impact of patients’ detailed nutritional statuses on morbidity and survival outcomes in patients with locally advanced rectal cancer undergoing combined neoadjuvant treatment followed by surgery is still rarely assessed. This study aimed to evaluate patients body composition indices in association with morbidity and oncological outcomes. However, malnutrition is often misunderstood as a situation related to a low BMI and is only defined according to the weight and height of the patient. In the present study, neither a low nor a high BMI was associated with increased morbidity, a higher rate of AL, or a worse oncological outcome. Therefore, the association between BMI and body composition indices was assessed in this study. On the one hand, visceral adiposity (*p* < 0.01), subcutaneous adiposity (*p* < 0.01), and total fat rate (*p* = 0.001) were significantly associated with an increased BMI, and on the other hand, those indices were not related to increased morbidity or worse oncological outcomes.

Sarcopenia, or sarcopenic obesity, is more likely to be associated with increased morbidity and oncological outcome than overweight or underweight alone. Sarcopenia is a syndrome characterized by the progressive and generalized loss of skeletal muscle mass and strength, carrying risks of physical disability, poor quality of life, and death [29]. However, sarcopenia is not only seen in older patients; it may be associated with conditions that are not exclusively seen in older people, such as cancer-related malnutrition or limited physical activity [29]. Several mechanisms are known to be involved in the processes of malnutrition and muscle wasting. These mechanisms involve, among others, protein synthesis, proteolysis, muscle fat content, and neuromuscular integrity, as seen with the metabolic changes in cancer patients [30]. In this study, we were able to clearly demonstrate that older patients and patients with reduced skeletal muscle areas have a significantly increased risk of complications. As shown previously, overweight and underweight alone have no association with impaired morbidity. This study confirmed the findings of previous studies which have identified the association between muscle wasting and increased overall morbidity, higher anastomotic leakage rates, and worsened oncological outcomes [29,31,32]. However, anastomotic leakage was not found to be affected by body composition indices or age in this study. Patients with an increased CCI had a significantly higher risk for AL; 50.0% of the patients included into this study had one or more comorbidities, respectively. These data underline the risk of comorbidities for AL, but do not automatically indicate a worse long-term oncological outcome. 

However, sarcopenic obesity seems to play a substantial role in the deterioration of the oncological outcome in locally advanced rectal cancer patients. Sarcopenic obesity is characterized by a combination of high body fat and low muscle function accompanied by low skeletal mass [33]. It has long been assumed that age-related loss of weight, along with loss of muscle mass, is mainly responsible for weakness in older people [30]. Moreover, changes in muscle composition and fat infiltration lower muscle quality and function [34]. In conditions such as malignancy, lean body mass is lost, while fat mass may be preserved [30]. The association between sarcopenic obesity and adverse oncological outcomes remains unclear. Systemic inflammation might be a possible explanation [35]. A systemic inflammatory condition is known to increase the risk of cancer and reduce the response of therapy [36,37]. Sarcopenia and systemic inflammation are known to be concordant, and would be substantial in obese patients who experiencing sarcopenic obesity [37,38]. These immunologic reactions may explain why patients with sarcopenic obesity had significantly worse DFS and OS in this cohort (Figure 2). No other body composition indices were found to be associated with a worsened oncological outcome, which is in accordance with previous published literature [37,39]. We assessed the effect of sarcopenic obesity on DFS using two different statistical approaches. Our Kaplan–Meier survival analysis demonstrated a significant difference in DFS between the intervention and control groups (log-rank test: *p* = 0.04). However, the chi-squared test comparing DFS status (yes or no) between the groups did not yield a significant result (*p* = 1), indicating that the association between patients with and without sarcopenic obesity and the binary outcome of DFS is not statistically significant. The discrepancy between the two tests may be attributed to their differing sensitivities and underlying assumptions. While the Kaplan–Meier analysis captured the time-to-event aspect of the data, the chi-squared test focused solely on the categorical outcome. 

This study has some limitations to be mentioned. First, this study is of a retrospective design, which implies selection bias. Second, as we analyzed the effects of body composition indices prior to treatment, we were unable to examine postoperative changes over the follow-up time. Third, this is a single-center study with central European patients. Because of nutritional differences, this study may be not applicable worldwide. Furthermore, a considerable number of participants had to be excluded because of unavailable CT scans. Finally, a relatively small number of included patients implicates a statistical bias. Nevertheless, this study clearly demonstrated that sarcopenic obesity is a risk factor for worse oncological outcome. Adequate nutritional therapy intervention along with physical activity both prior to and during neoadjuvant therapy is essential in order to counteract sarcopenic obesity and may alter the DFS and OS. This study is—to our knowledge—the first study to focus on the pretherapeutic conditions of patients with locally advanced rectal cancer who underwent multimodal therapy in association with short- and long-term oncological outcomes.

## 5. Conclusions

Muscle waste, older age, and comorbidities were demonstrated as strong risk factors for increased overall morbidity. Sarcopenic obesity was associated with a worsened DFS and OS. This study underlines that BMI alone as risk factor for increased morbidity in patients undergoing curative therapy is obsolete. The role of nutrition and physical activity prior to medical therapy are crucial. 

## Figures and Tables

**Figure 1 nutrients-15-02632-f001:**
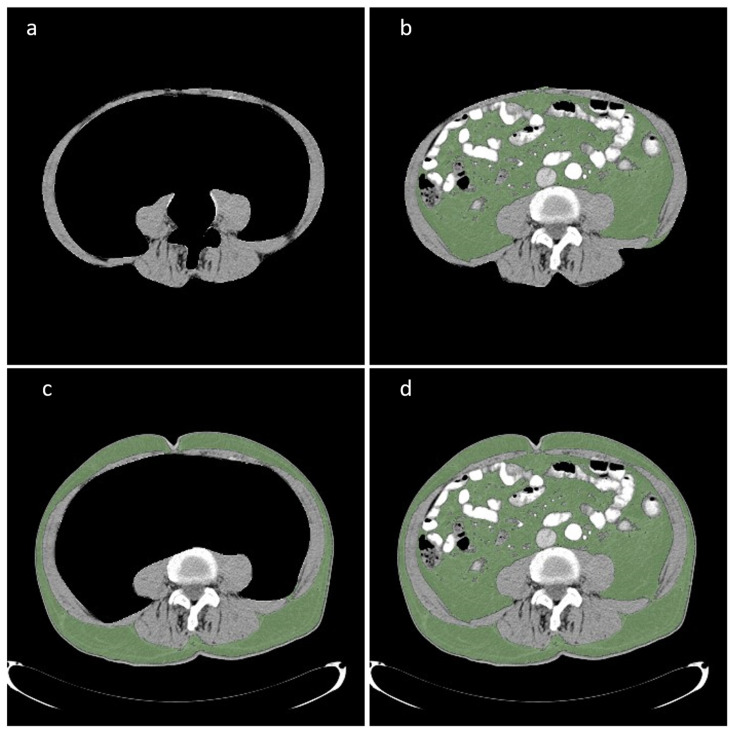
Evaluation of body composition indices with Horos^TM^, v3.3.6. (**a**) Muscle mass, (**b**) visceral fat, (**c**) subcutaneous fat, (**d**) total fat.

**Figure 2 nutrients-15-02632-f002:**
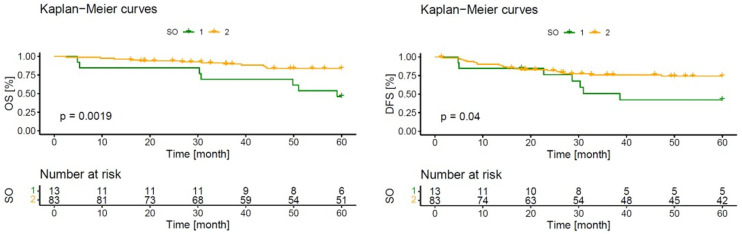
Kaplan–Meier survival curves of patients with (green line) and without (yellow line) sarcopenic obesity.

**Table 1 nutrients-15-02632-t001:** Patients’ characteristics.

Patients’ Characteristics	Total (*n* = 96)
Sex, Male/Female, *n* (%)	64 (66.7%)/32 (33.3%)
Age (year), mean ± std	64.0 ± 11.0
BMI (kg/m^2^), mean ± std	26.7 ± 4.2
ASA Classification, *n* (%)	
I	14 (14.6%)
II	49 (51.0%)
III	30 (31.3%)
IV	1 (1.0%)
V	0 (0.0%)
Tumor localization (in cm from the anal verge), mean ± std	7.5 ± 3.7
CEA level preoperative (µg/l), mean ± std	3.8 ± 4.4
Comorbidities, *n* (%)	49 (51.0%)
Charlson Comorbidity Index	5.4 ± 2.5
Preoperative therapy, *n* (%)	
Combined long-term chemo-radiation	80 (83.3%)
Short-term radiation (5 × 5 Gy)	16 (16.7%)
Type of Anastomosis, *n* (%) (*n* = 97)	
Stapled E-E	62 (64.6%)
Stapled S-E	24 (25.0%)
Hand sewn colo-anal anastomosis	10 (10.4%)
Operative technique, *n* (%)	
open	56 (58.3%)
laparoscopic	40 (41.8%)
Operation time (min), mean ± std:	207.9 ± 64.6
Complications, *n* (%)	36 (37.5%)
Anastomotic leakage	20 (20.8%)
Superficial SSI	1 (1.0%)
Bleeding	2 (2.1%)
Bowel obstruction	5 (5.2%)
Renal failure	1 (1.0%)
Stoma obstruction	1 (1.0%)
Parastomal hernia	2 (2.1%)
Others	3 (3.1%)
Clavien-Dindo Classificatio, *n* (%)	
I	2 (2.1%)
II	4 (4.2%)
III	28 (29.2%)
IV	0 (0.0%)
V	1 (1.0%)
Duration of hospital stay (d), mean ± std	20.0 ± 14.2
TME quality, *n* (%)	
Grade 1 (good)	67 (69.8%)
Grade 2 (moderate)	21 (21.9%)
Grade 3 (poor)	8 (8.3%)
Pathological yT stages, *n* (%)	
0	13 (13.5%)
Tis	2 (2.1%)
T1	3 (3.1%)
T2	33 (34.4%)
T3	43 (44.8%)
T4	2 (2.1%)
Pathological yN Stage, *n* (%)	
N0	64 (66.7%)
N1	19 (19.8%)
N2	13 (13.5%)
Postoperative UICC-Stage, *n* (%)	
0	14 (14.6%)
I	28 (29.2%)
II	22 (22.9%)
III	32 (33.3%)
Residual Tumor (R1), *n* (%)	4 (4.2%)
Adjuvant chemotherapy, *n* (%)	47 (49.0%)
Follow-up:	
Follow-up time (months), mean ± std	69.0 ± 46.5
Local recurrence, *n* (%)	6 (6.3%)
Distant metastasis, *n* (%)	22 (22.9%)

Values are given in mean ± standard deviation or as numbers and percentages. Abbreviations: BMI = Body mass index, ASA = American Society of Anesthesiologists, CCI = Charlson Comorbidty Index, UICC = Union for International Cancer Control, SSI = Surgical side infection, TME = Total mesorectal excision.

**Table 2 nutrients-15-02632-t002:** Association between BMI and body composition indexes.

		Normal	Underweight	Overweight	*p*-Value
Subcutaneous adiposity					<0.01
	No	23 (76.7%)	4 (100.0%)	24 (38.7%)	
	Yes	7 (23.3%)	0 (0.0%)	38 (61.3%)	
Visceral adiposity					<0.01
	No	15 (50.0%)	4 (100.0%)	13 (21.0%)	
	Yes	15 (50.0)	0 (0.0%)	49 (79.0%)	
Visceral to total fat ratio					0.058
	No	14 (46.7%)	2 (50.0%)	16 (25.8%)	
	Yes	16 (53.3%)	2 (50.0%)	46 (74.2%)	
Sarcopenic obesety					0.771
	No	3 (10.0%)	0 (0.0%)	7 (11.3%)	
	Yes	27 (90.0%)	4 (100.0%)	55 (88.7%)	
Skelettal muscle index					0.138
	No	5 (16.7%)	2 (50.0%)	8 (12.9%)	
	Yes	25 (83.3%)	2 (50.0%)	54 (87.1%)	
Subcutaneous to total fat ratio					0.06
	No	12 (40.0%)	4 (100.0%)	34 (54.8%)	
	Yes	18 (60.0%)	0 (0.0%)	28 (45.2%)	
Skelettal muscle area					0.157
	No	7 (23.3%)	2 (50.0%)	9 (14.5%)	
	Yes	23 (76.7%)	2 (50.0%)	53 (85.5%)	
Total fat					0.001
	No	27 (90.0%)	4 (100.0%)	33 (53.2%)	
	Yes	3 (10.0%)	0 (0.0%)	29 (46.8%)	

Values are given in numbers and percentages. Abbreviations: SMA = Scelettal muscle area, VF = Visceral fat.

**Table 3 nutrients-15-02632-t003:** Association between baseline characteristics. surgical characteristics. therapy characteristics and nutrition indexes.

Variable	Overall Morbidity			Anastomotic Leakage			Disease Free Survival			Local Recurrency		
	yes	no	*p*-value	yes	no	*p*-value	yes	no	*p*-value	yes	no	*p*-value
Age	69.0 ± 10.0	62.0 ± 11.0	0.004	68.0 ± 10.0	64.0 ± 12.0	0.151	65.0 ± 11.0	64.0 ± 12.0	0.829	71.0 ± 12.0	64.0 ± 11.0	0.178
Sex (Male/Female)	23 (65.7%)/12 (34.3%)	41 (67.2%)/20 (32.8%)	1	16 (69.6%) /7 (30.4%)	48 (65.8%)/25 (34.3%)	0.933	47 (64.4%)/26 (35.6%)	17 (73.9%)/6 (26.1%)	0.554	4 (66.7%) /2 (33.3%)	60 (66.7%)/30 (33.3%)	1
CCI	6.6 ± 2.5	4.66 ± 2.2	<0.01	6.6 ± 2.5	5.0 ± 2.4	0.006	5.4 ± 2.5	5.39 ± 2.5	0.89	4.8 ± 0.9	5.4 ± 2.6	0.909
Type of neoadjuvant therapy			0.129			0.285			0.134			0.572
long-term	26 (74.3%)	54 (88.5%)		17 (73.9%)	63 (86.3%)		58 (79.5%)	22 (95.7%)		6 (100.0%)	74 (82.2%)	
short-term	9 (25.7%)	7 (11.5%)		6 (26.1%)	10 (13.7%)		15 (20.5%)	1 (4.3%)		0 (0.0%)	16 (17.8%)	
Type of surgery			0.971			0.968			0.968			1
open	21 (60.0%)	35 (57.4%)		14 (60.9%)	42 (57.5%)		42 (57.5%)	14 (60.9%)		4 (66.7%)	52 (57.8%)	
laparoscopic	14 (40.0%)	26 (42.6%)		9 (39.1%)	31 (42.5%)		31 (42.5%)	9 (39.1%)		2 (33.3%)	38 (42.2%)	
TME-Quality			0.695			0.997			0.639			0.733
1 (good)	26 (74.3%)	41 (67.2%)		16 (69.6%)	51 (69.9%)		52 (71.2%)	15 (65.3%)		4 (66.7%)	63 (70.0%)	
2 (moderate)	6 (17.1%)	15 (24.6%)		5 (21.7%)	16 (21.9%)		16 (21.9%)	5 (21.7%)		1 (16.7%)	20 (22.2%)	
3 (worse)	3 (8.6%)	5 (8.2%)		2 (8.7%)	6 (8.2%)		5 (6.9%)	3 (13.0%)		1 (16.7%)	7 (7.8%)	
BMI			0.72			0.635			0.995			0.591
Normal	12 (34.3%)	18 (29.5%)		9 (39.1%)	21 (28.8%)		23 (31.5%)	7 (30.4%)		1 (16.7%)	29 (32.2%)	
Underweight	2 (5.7%)	2 (3.3%)		1 (4.4%)	3 (4.1%)		3 (4.1%)	1 (4.4%)		0 (0.0%)	4 (4.4%)	
Overweight	21 (60.0%)	41 (67.2%)		13 (56.5%)	49 (67.1%)		47 (64.4%)	15 (65.2%)		5 (83.3%)	57 (63.3%)	
Subcutaneous adiposity	250.4 ± 97.2	225.3 ± 103.1	0.146	254.6 ± 85.4	228.2 ± 105.5	0.141	244.3 ± 101.7	203.4 ± 95.1	0.108	232.8 ± 66.0	234.6 ± 103.6	0.862
Visceral adiposity	165.0 ± 71.4	177.0 ± 91.7	0.787	171.0 ± 56.8	173.2 ± 92.2	0.874	175.0 ± 86.0	165.2 ± 81.6	0.631	206.3 ± 85.1	170.4 ± 84.6	0.454
Skelettal muscle index	45.1 ± 15.9	51.4 ± 13.5	0.053	45.6 ± 12.2	50.2 ± 15.3	0.135	49.4 ± 14.3	48.2 ± 16.0	0.874	43.3 ± 13.9	49.5 ± 14.7	0.329
Subcutaneous to total fat ratio	0.6 ± 0.1	0.6 ± 0.1	0.096	0.6 ± 0.1	0.6 ± 0.1	0.363	0.6 ± 0.1	0.6 ± 0.1	0.178	0.5 ± 0.1	0.6 ± 0.1	0.345
Skelettal muscle area	129.4 ± 47.6	151.0 ± 46.3	0.045	131.6 ± 38.2	146.8 ± 50.1	0.152	145.0 ± 48.3	137.1 ± 46.3	0.609	122.5 ± 42.7	144.5 ± 48.0	0.217
Total fat	412.6 ± 149.6	402.4 ± 161.3	0.615	421.3 ± 123.5	401.3 ± 166.2	0.474	417.9 ± 155.1	368.6 ± 158.3	0.142	439.2 ± 142.4	403.9 ± 157.9	0.7
Visceral to total fat ratio	0.4 ± 0.1	0.4 ± 0.1	0.204	0.42 ± 0.13	0.4 ± 0.1	0.699	0.4 ± 0.1	0.5 ± 0.1	0.243	0.5 ± 0.1	0.4 ± 0.1	0.393
Sarcopenic obesity	8 (22.9%)	3 (4.9%)	0.02	3 (13.0%)	8 (11.0%)	1	9 (12.3%)	3 (13.0%)	1	0 (0.0%)	11 (12.2%)	0.804

Values are given as mean ± standardization or as numbers and percentages. Abbreviations: CCI = Charlson Comorbidity Index, TME = Total mesorectal excision, BMI = Body mass index, SMA = Scelettal muscle area, VF = Visceral fat.

## Data Availability

The datasets generated and/or analyzed during the current study are available from the corresponding author upon reasonable request.

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
