# Peer review of "Sarcopenic Obesity Is a Risk Factor for Worse Oncological Long-Term Outcome in Locally Advanced Rectal Cancer Patients: A Retrospective Single-Center Cohort Study"

_nutrients, 2023, doi:10.3390/nu15112632_

Round 1

Reviewer 1 Report

Thank you for the opportunity to review this work. The research topic is important and this evidence will help support the work of many other researchers in this field. Overall, the articles talks about the muscle and fat mass, and how together (sarcopenic obesity) they can impact long term outcomes in advanced rectal cancer patients. It is important to improve the practice of pre-operative nutritional assessment beyond BMI.

Grammar and English – please correct the various grammar errors throughout the paper. There are also a fair bit of missing punctuations. Sentence structure issues too. There are also spelling errors e.g., Charlsen (should be Charlson) comorbidity index. Please send to an English editor to best represent the wonderful work that you have done.

Title

-          Please state that it is a retrospective cohort study and retitle.

Results

-          Please change to 3 significant numbers for the results. There is no additional benefits of using 2 decimal place. For instance, 49 patients (51.04%) – it is not helpful to have that additional 04 in those results. Instead, it makes it harder to read.

-          Please section these results and provide the tables and figures listed in the manuscript. They were not found in supplementary materials too. I cannot further review this section.

Discussion

-          Again, I strongly recommend an English editor. Malnutrition and sarcopenia do not enhance morbidity.

-          I agree that better pre-operative and post-operative nutritional assessment and treatment is needed.

-          May I know what is a fate rate on line 188?

-          Please replace the word elderly with older adults.

-          This evidence from this study can only suggest a direction, albeit a strong one.

-          While I am glad that the authors shared evidence on this topic, I think that they are overstating the results from just one study. Afterall, this is not a meta-analysis.

English editor needed.

Author Response

Dear Reviewer!

Thank you very much for reviewing this manuscript and for all your recommendations. The manuscript was presented to a native English speaker, as suggested. All corrections have been marked and highlighted. I hope we have answered all your queries correctly to improve this manuscript. Please don't hesitate to contact me if you have further questions or suggestions.

Kind regards

P. Tschann

Point for point answer:

Grammar and English – please correct the various grammar errors throughout the paper. There are also a fair bit of missing punctuations. Sentence structure issues too. There are also spelling errors e.g., Charlsen (should be Charlson) comorbidity index. Please send to an English editor to best represent the wonderful work that you have done.

The manuscript was sent to a native English speaker. All corrections have been highlighted.

Title

-          Please state that it is a retrospective cohort study and retitle.

Title was changed.

Results

-          Please change to 3 significant numbers for the results. There is no additional benefits of using 2 decimal place. For instance, 49 patients (51.04%) – it is not helpful to have that additional 04 in those results. Instead, it makes it harder to read.

Corrected in all tables and in the main text. Changes are highlighted in yellow.

-          Please section these results and provide the tables and figures listed in the manuscript. They were not found in supplementary materials too. I cannot further review this section.

Thank you for this commentary. I uploaded all tables and figures in the first version. I’ll try again. Furthermore, I corrected all tables to three significant numbers, as suggested before. Please do not hesitate to contact me if it doesn’t work. Thank you!

Discussion

-          Again, I strongly recommend an English editor. Malnutrition and sarcopenia do not enhance morbidity.

English editing was performed, as suggested.  

-          May I know what is a fate rate on line 188?

Total fat rate. Corrected and highlighted.

-          Please replace the word elderly with older adults.

Corrected and highlighted.

-          This evidence from this study can only suggest a direction, albeit a strong one.

You are right. A sentence about the relatively low number of patients was added in the limitations section and highlighted in yellow.

-          While I am glad that the authors shared evidence on this topic, I think that they are overstating the results from just one study. Afterall, this is not a meta-analysis.

We added one previously published study, which also had a relatively low number of included patients. The trend is clear in the literature. To our knowledge, a meta-analysis on this topic does not currently exist. However, the number of published manuscripts on sarcopenia and its association with long-term outcomes is rapidly increasing on the one hand. On the other hand, the association between other body composition indexes and long-term outcomes is rarely described in the previous published literature.

Reviewer 2 Report

This study investigated the impact of several measures for body composition on outcomes in patients with advanced rectal cancer. Patients were included consecutively. The inclusion period is given. Statistical methods are presented.

Please check, if all patients who fulfilled the inclusion criteria could be included. It might be that some patients denied to participate.

Furthermore, please add statements on the sample size calculation and on the power of your results. 

were that were However, 

Author Response

Dear Reviewer!

Thank you very much for reviewing this manuscript and for all your recommendations. All corrections are marked and highlighted. I hope we answered all your queries correctly to improve this manuscript. Please don’t hesitate to contact me if you have further questions/suggestions.

Kind regards

P. Tschann

Point for point answer:

This study investigated the impact of several measures for body composition on outcomes in patients with advanced rectal cancer. Patients were included consecutively. The inclusion period is given. Statistical methods are presented.

Please check, if all patients who fulfilled the inclusion criteria could be included. It might be that some patients denied to participate.

Thank you very much for your comments. The study design was retrospective, and the ethical board approved the study without any additional comments. All patients who met the inclusion criteria were included in this study, and the CT scans that were performed before therapy were evaluated.

Furthermore, please add statements on the sample size calculation and on the power of your results. 

Thank you for this advice. We added a statement in the results section (P5,6 L142-145).